# Nondiffracting supertoroidal pulses and optical "Kármán vortex streets"

Yijie Shen [1,2] ✉, Nikitas Papasimakis [3] & Nikolay I. Zheludev [1,3]

Supertoroidal light pulses, as space-time nonseparable electromagnetic waves, exhibit unique topological properties including skyrmionic configurations, fractal-like singularities, and energy backflow in free space, which however do not survive upon propagation. Here, we introduce the non-diffracting supertoroidal pulses (NDSTPs) with propagation-robust skyrmionic and vortex field configurations that persists over arbitrary propagation distances. Intriguingly, the field structure of NDSTPs has a similarity with the von Kármán vortex street, a pattern of swirling vortices in fluid and gas dynamics with staggered singularities that can stably propagate forward. NDSTPs will be of interest as directed channels for information and energy transfer applications.

The topological properties of light have been a subject of fascination and intense research interest over the last half century[1–3] with implications for light-matter interactions[4–6], nonlinear physics[7–9], spin-orbit coupling[10–12], microscopy and imaging[13–15], metrology[16], and information transfer[17–22]. Light pulses can be simultaneously structured in the space and time domains[23–25]. Three-dimensional topological structures have also been studied recently, such as the toroidal phase vortices in scalar light pulses[26–28] and the photonic skyrmions and hopfions with particle-like topologies in vector beams[29–31]. As a special type of topologically structured light with both ultrafast temporal and vectorial electromagnetic configurations, the toroidal light pulses are of particular interest[32], as they can engage toroidal excitations in matter[33,34] and exhibit space-time nonseparability and isodiffraction[35–37]. A generalization of toroidal pulses, the supertoroidal pulses (STPs) exhibit striking topology including fractal-like singularities, vortex rings, energy backflows, and optical skyrmionic patterns[38,39]. To date, all known electromagnetic analogs of skyrmions are short-lived and do not persist upon propagation.

In this Letter, we introduce the nondiffracting supertoroidal pulses (NDSTPs), an extended family of STPs that propagate without diffraction. NDSTPs exhibit the topological properties of STPs, such as energy backflow, fractal organization of singularities, and skyrmion-like field configurations. In contrast to previously considered forms of skyrmionic light, typically observed in the steady state[40–42], NDSTPs exhibit spatiotemporal topological features that persist upon propagation. Intriguingly, the structure of an NDSTP resembles that of Kármán vortex street (KVS), propagating staggered vortex arrays observed in fluid and gas dynamics[43], previously observed in continuous-wave structured light fields[44–47].

## Results

### Nondiffracting supertoroidal pulses

(ND)STPs originate from the "electromagnetic directed-energy pulse trains" (EDEPTs) theory introduced by Ziolkowski to obtain focused few-cycle electromagnetic pulses that are localized finite-energy and space-time non-separable solutions of Maxwell's equations[48]. As such, NDSTPs are also finite energy space-time non-separable pulses. The first step is finding a scalar generating function $f(\mathbf{r}, t)$ that satisfies Helmholtz's scalar wave equation:

$$\left(\nabla^2 - \frac{1}{c^2}\frac{\partial^2}{\partial t^2}\right)f(\mathbf{r},t) = 0, \tag{1}$$

where $\mathbf{r} = (r, \theta, z)$ represents the position vector (in a cylindrical coordinate system), $t$ is time, $c = 1/\sqrt{\varepsilon\mu}$ is the speed of light, and the $\varepsilon$ and $\mu$ are the permittivity and permeability of medium. Here, we consider propagation in free-space and thus permittivity and permeability are set to their vacuum values. The exact solution of $f(\mathbf{r}, t)$ can be

[1]Centre for Disruptive Photonic Technologies, School of Physical and Mathematical Sciences & The Photonics Institute, Nanyang Technological University, Singapore 637378, Singapore. [2]School of Electrical and Electronic Engineering, Nanyang Technological University, Singapore 639798, Singapore. [3]Optoelectronics Research Centre & Centre for Photonic Metamaterials, University of Southampton, Southampton SO17 1BJ, UK. ✉e-mail: yijie.shen@ntu.edu.sg

given by the modified power spectrum method proposed by Ziolkowski[48,49]:

$$f(\mathbf{r},t) = f_0 \frac{e^{-s/q_3}}{(q_1 + i\tau)(s + q_2)^\alpha} \quad (2)$$

where $f_0$ is a normalizing constant, where $s = r^2/(q_1 + i\tau) - i\sigma$, $\tau = z - ct$, $\sigma = z + ct$, $q_1, q_2, q_3$ are real positive parameters with units of length, and the real dimensionless parameter $\alpha$ must satisfy $\alpha \geq 1$. The parameter $\alpha$ controls the energy confinement at focus and spatial divergence of the pulse. In particular, the pulse carries infinite energy when $\alpha < 1$, the pulse has localized finite energy when $\alpha \geq 1$, and with increasing $\alpha$ the pulse becomes more divergent upon propagation, therefore the value of parameter $\alpha$ is usually fixed at unity to ensure finite energy of the pulse[48,50]. In order to retrieve the toroidal electromagnetic fields, we construct the Hertz potential as $\mathbf{\Pi} = \nabla \times \hat{\mathbf{z}} f(\mathbf{r},t)$, then the fields of transverse electric (TE) mode, i.e. the electric field is azimuthal ($E_\theta$) and the magnetic field is nontransverse (including $H_r$ and $H_z$ components), can be solved by[50] (see Supplementary Note 1 for more detailed derivations):

$$\mathbf{E}(\mathbf{r},t) = -\mu_0 \frac{\partial}{\partial t} \nabla \times \mathbf{\Pi} \quad (3)$$

$$\mathbf{H}(\mathbf{r},t) = \nabla \times (\nabla \times \mathbf{\Pi}) \quad (4)$$

For the previous solution of supertoroidal pulses ($q_3 = \infty$), $q_1$ is the effective wavelength as a constant proportional to the central wavelength $\lambda$ of the pulse ($q_1 \approx 0.24\ \lambda$), $q_2$ is related to the Rayleigh length ($q_2 = z_0/2$) describing the spatial divergence along longitudinal direction[39]. While, when $q_3$ takes finite values, it quantifies the spatial divergence along the transverse direction of the structured pulse at focus, which refers to the distance from the pulse center along the transverse direction where the amplitude profile becomes bifurcate. At the same time, the longitudinal divergence of the pulse has a trend to be weakened becomes weaker and it is, no longer described by $q_2$, while the pulse becomes nondiffracting.

A representative elementary toroidal light pulse is shown in Fig. 1a. The transverse magnetic (TM) mode can be obtained by exchanging electric and magnetic fields of the TE mode. In prior works, the conditions $q_1 \ll q_2$, $\alpha = 1$, and $q_3 \to \infty$ were assumed to generate various focused structured pulses[50–54], where $q_1$ and $q_2$ determine the wavelength and the longitudinal divergence or Rayleigh length of the pulses, respectively, as marked in Fig. 1a. The condition $q_2 \gg q_1$ allows to avoid pathologies related to the presence of backward propagating components[55,56]. The case of $\alpha \geq 1$ and $q_3 \to \infty$ was recently studied leading to the introduction of supertoroidal pulses[39]. However, the solutions with finite values of $q_3$ have never been studied before. In this work, we explore STPs with finite $q_3$ (see Supplementary Note 1 for derivations). A characteristic example of such a pulse with $q_3 = q_1$ and $\alpha = 1$ is plotted in Fig. 1b. Here, $q_3$ defines the degree of transverse divergence: with decreasing value of $q_3$ the pulse envelope is gradually squeezed into a dumbbell-like shape and eventually becomes nondiffracting for $q_3 = q_1$ and $\alpha = 1$, approaching the case of nondiffraction $|E(r, \theta, z, t)| = |E(r, \theta, z + \Delta z, t + \Delta z/\nu)|$ ($\Delta z$ is a given propagation distance and $\nu$ is group velocity)[57], see Supplementary Note 2. Note that, exact nondiffracting waves like ideal Bessel beams or plane waves only exist in theory and cannot be realized experimentally as practical light pulses would carry infinite energy. Here, we use the term "nondiffracting" to describe finite energy waves that propagate without diffraction over very large (but finite) distances.

The evolution of the pulse from diffracting to nondiffracting is illustrated in Fig. 2. An intermediate case of a weakly-diffracting supertoroidal pulse in terms of $q_1$, $q_2$, and $q_3$, is presented Fig. 2a. To

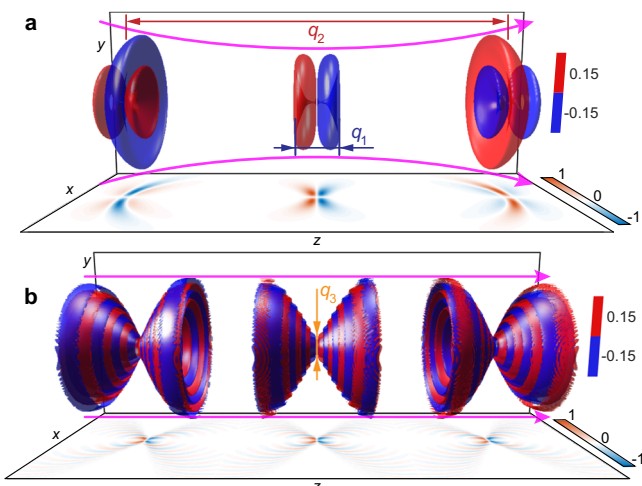

**Fig. 1 | Spatiotemporal topological pulse evolutions.** Propagation evolution of an elementary toroidal pulse (**a**) and a NDSTP (**b**): The distributions of the normalized TE mode electric fields of the pulses at three various times $t = 0, \pm q_2/(2c)$ are plotted on the $x$-$z$ plane. The red and blue 3D isosurfaces represent the locations with electric field amplitude equal to 0.15 and −0.15, respectively, normalized to the corresponding maximum value. The blue, red, and yellow arrows mark the lengths of optical cycle ($q_1$), Rayleigh range ($q_2$), and transverse divergence ($q_3$), respectively. Parameters used in simulation: **a** $q_2/q_1 = 40$; **b** $q_2/q_1 = 100$ and $q_3/q_1 = 1$. Purple arrows mark the propagation profile. See Supplementary Movie 1 for the dynamic evolutions.

illustrate the dependence of the (non)diffraction of the pulse on $q_3$, we calculate the $z$-dependent full width at half maximum (FWHM) radius of the transverse intensity pattern of the STPs with $q_2 = 100q_1$ and $q_3$ varying from infinity to $q_1$, for propagation distances from focus to $z = 10^3 q_1$, see Fig. 2b. In the extreme case of $q_3 \to \infty$ (fundamental toroidal light pulse), the FWHM of the pulse follows a hyperbolic trajectory similar to a focused Gaussian beam. With decreasing $q_3$, the divergence becomes weaker and the pulse approaches a nondiffracting state for $q_3 \leq 5q_1$. Figure 2c–f shows the spatiotemporal evolution upon propagation for toroidal pulses with different $q_3$ values. Here, decreasing $q_3$ results in a faster spatiotemporal evolution of the cycle structure of the pulse (see Supplementary Movie 1), due to its increasingly complex shape. When the $q_3$ value is decreased further to $q_1$, the pulse becomes X-shaped (Fig. 1b), which reveals a conical structure. We note that $q_1$ is independent of $q_3$ and thus we would not expect a change in the former when the latter varies. The counterintuitive decrease in beam waist with a decrease in divergence angle can be attributed to the presence of "long thick wings" at the peripheral area of the pulse. Upon propagation, the cycle structure keeps evolving on the conical surface akin to a breather (see Supplementary Movie 1). Note that similar X-type nondiffracting pulses have been considered previously both theoretically and experimentally and are typically termed Bessel-X pulses[58–60]. However, previous works focused on scalar long-pulses within the slowly-varying amplitude envelope approximation. Here, our NDSTPs are few-cycle, space-time nonseparable with nontrivial electromagnetic toroidal topology lacking in the Bessel-X pulses. Table 1 summarizes the parameter requirements for various cases of fundamental toroidal pulse, STP, NDSTP, and intermediate states.

## Singularities and topological properties

The emergence of NDSTPs allows us to explore intriguing topological optical effects. In our recent work, we showed that supertoroidal pulses exhibit a complex topology (controlled by the parameter $\alpha$), including self-similar fractal-like patterns, matryoshka-like singularity arrays, skyrmions, and areas of energy backflow[39]. Here, we show that

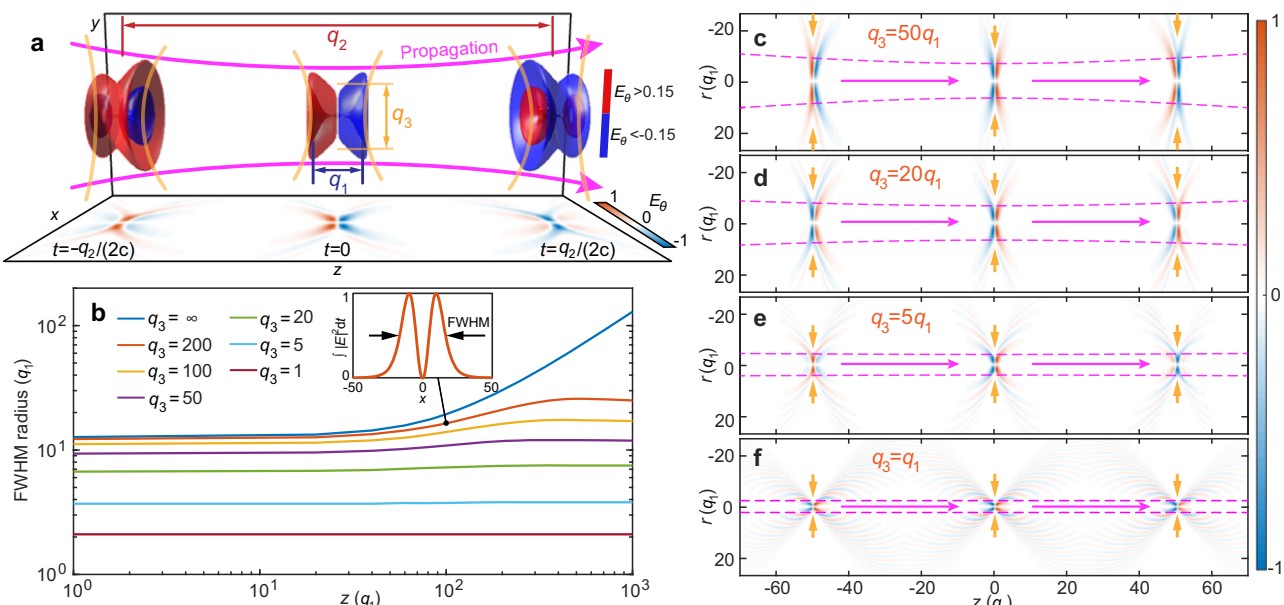

**Fig. 2 | Nondiffracting propagation properties.** **a** Propagation evolution of a transversely divergent pulse with $q_3/q_1 = 20$: The distributions of the normalized electric fields of the pulses at three various times $t = 0, \pm q_2/(2c)$ are plotted on the $x$-$z$ plane. The red and blue 3D isosurfaces represent the locations with electric field amplitude equal to 0.15 and −0.15, respectively, normalized to the corresponding maximum value. The presented case lies between the fundamental toroidal pulse [$q_3 \to \infty$; see Fig. 1a] and a NDSTP [$q_3 = q_1$; see Fig. 1b]. The transition from the former to the latter can be seen in Supplementary Movie 1. **b** The evolution of pulse width in the transverse upon propagation for different $q_3$ values, where the numerically calculated divergence was marked. The inset shows a 1D cross-section of the pulse intensity in the transverse plane with the pulse width quantified by the full width at half maximum (FWHM). **c**–**f** The spatiotemporal evolution of the pulses with various $q_3/q_1$ values of 50, 20, 5, and 1, respectively. In each panel, the purple dashed lines mark the FWHM of the corresponding pulses. $q_1$ is equal to unity for all case.

such topological structures are also present in NDSTPs. However, whereas in the former case, the self-similar topological structures were only instantaneously observed at focus (within a region of length $q_2$), in the NDSTP, they can be observed over arbitrarily long distances (see example in Fig. 3b). For instance, we have shown in our previous work that STPs exhibit a complex topology that evolves rapidly upon pulse propagation with elements of self-similarity consisting of concentric matryoshka-like spherical shells over which the electric field vanishes (see Fig. 3a). Similar propagation-robust fractal-like topological structures exist in the NDSTP (see Fig. 3b) and persist upon pulse propagation (see also Supplementary Movie 2).

The complex topology of the NDSTPs manifests also in the magnetic field distribution. Figure 4a shows the magnetic field distribution of a NDSTP with $q_2/q_1 = 100$ and $q_3/q_1 = 1$ at $t = 0$. The magnetic field includes both radial and longitudinal components resulting in vector singularities of vortex and saddle types. The saddle-type singularities are distributed along the propagation axis, while the vortices trace an off-axis trajectory, see Fig. 4a1. Such a singularity distribution induces multiple electromagnetic skyrmions at transverse planes of the pulse, see Fig. 4a2. Particularly, the optical skyrmion texture holds deep-subwavelength features in its field vector reversal regions, for instance shown in the plot of absolute value of normalized $H_z$ field as the insert of Fig. 4a2, where the first reversal region occurs within size of $q_1/2$ and the second of $q_2/10$ along the radial direction (evaluated by full width at half maximum). In contrast to the previous optical skyrmions in free

space that do not propagate[41,42] or only exist around focus and collapse rapidly upon propagation[39], here in the NDSTP, the skyrmions persist upon propagation with their topological texture exhibiting a periodic behaviour alternating between four different skyrmion types (combination of opposite two polarities and two helical angles), see Supplementary Movie 3 and Supplementary Note 3. The evolution of skyrmion or toroidal structures is coupled with the evolution of optical cycles due to the phase shift along the $z$-axis. There are only two cases with skyrmion numbers of ±1 that can be observed in the transverse plane, and the total skyrmion number is zero due to the symmetry of the pulse structure.

Figure 4b shows the Poynting vector field distribution of the NDSTP, which possesses layered energy forward-flow and backflow structures (Fig. 4b1). Interestingly, energy transport is mediated by the vortex arrays of the magnetic field. Vortices on the front half of the pulse act as energy sources, whereas the vortices in the rear half behave as sinks. In between sources and sinks, we observe areas of extended backflow. In areas of high intensity, energy flows primarily forward, ensuring the propagation of the pulse.

### Optical analogy of Kármán Vortex Streets (KVS)

Moreover, we observe that the vortex arrays exhibited by the NDSTPs form a striking trail of two-vortex clusters (with opposite circulations, namely a vortex dipole) propagating in a periodic staggered manner, evocative of the KVS structure. In fluid dynamics, a KVS is a classic pattern of swirling vortices caused by a nonlinear process of vortex shedding, which refers to the unsteady separation of the flow of fluid around blunt bodies. Vortex-street-like optical fields were reported previously in stationary optical fields exhibiting phase vortex patterns[44–47]. In contrast, here we observe KVS-like structures in propagating electromagnetic pulses in the linear regime. Due to the nondiffracting nature of NDSTPs, such structures persist upon propagation. In fluid dynamics, a KVS is a pattern of repeating swirling vortices constructed in the flow velocity field, whilst in our optical

## Table 1 | Classification of supertoroidal pulses

| Pulse type | Parameter requirement |
|---|---|
| Toroidal pulse | $q_1 \ll q_2$, $q_3 \to \infty$, $\alpha = 1$ |
| Supertoroidal pulse | $q_1 \ll q_2$, $q_3 \to \infty$, $\alpha > 1$ |
| Nondiffracting supertoroidal pulse | $q_1 \ll q_2$, $q_3 = q_1$, $\alpha = 1$ |
| Weakly-diffracting supertoroidal pulse | $q_1 \ll q_2$, $q_3 > q_1$, $\alpha > 1$ |

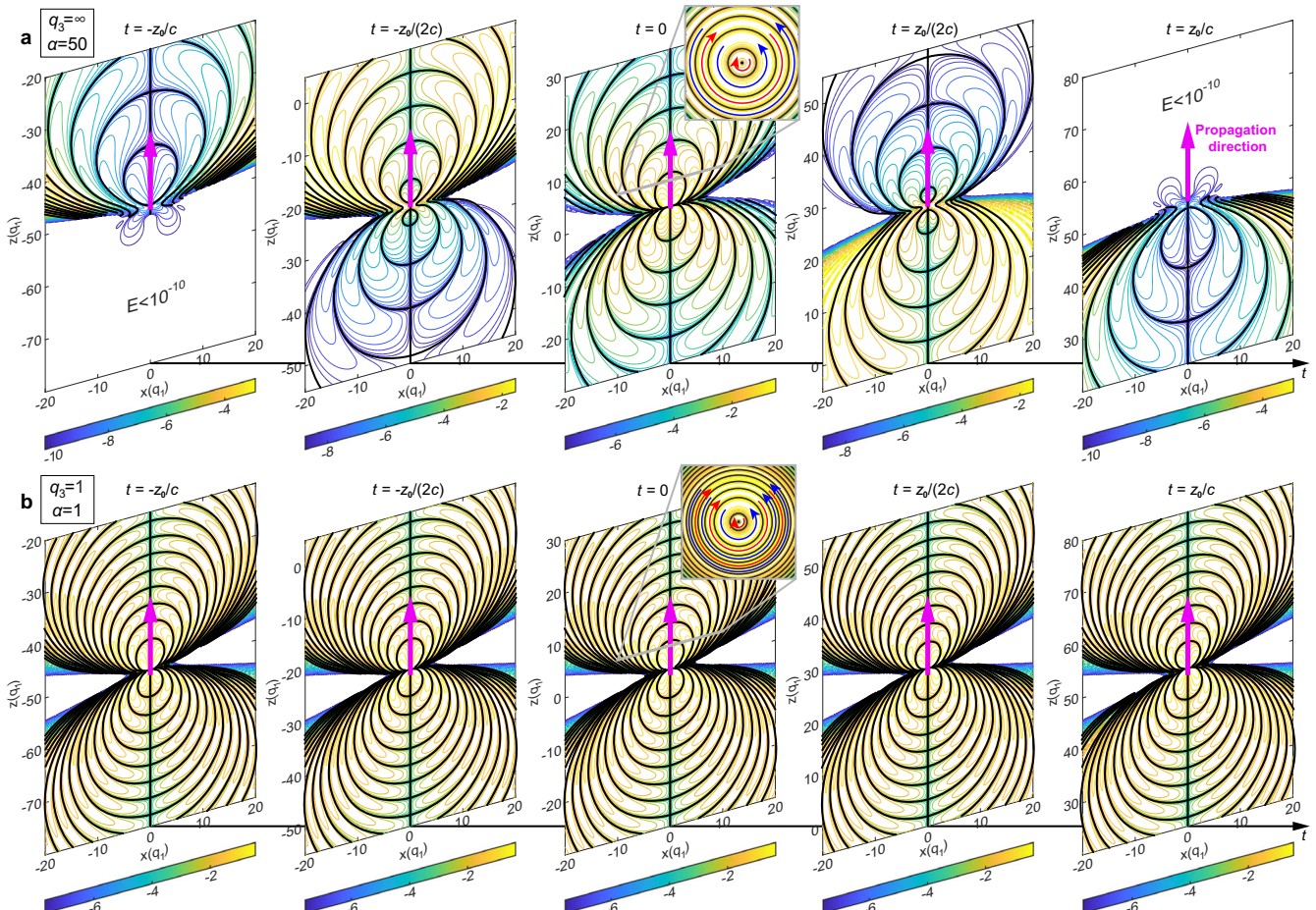

**Fig. 3 | Propagation robust singular structures.** The singular structures of electric field evolving with propagation versus the time for a focused STP ($q_3 = \infty$, $\alpha = 50$) (**a**) and a NDSTP ($q_3 = 1$, $\alpha = 1$) (**b**): each panel presents an isoline plot of the logarithm of the electric field ($\log_{10}|\vec{E}|$) of the pulse in the $x$-$z$ plane at a given time. The singularities are highlighted by bold black lines. Unit of length: $q_1$. See Supplementary Movie 2 for the dynamic evolutions of **a** and **b** other general cases with different parameters.

analogy, the KVS pattern is constructed by the electric or magnetic field of NDSTPs. Moreover, whereas KVS in fluid dynamics is a nonlinear phenomenon, here we show that similar effects can be observed in a linear system. The analogy between KVS in fluid flows and NDSTPs can be drawn further by considering for instance the motion of electrons along the vortex streets of a TM NDSTP or the propagation of supertoroidal pulses in nonlinear media. We note that KVS in fluid dynamics are typically 2D, while their optical analogs in NDSTPs comprise 3D sets of vortex rings, owing to the pulse cylindrical symmetry, see Fig. 5.

## Discussion

NDSTPs are nonpathological propagating solutions to Maxwell's equations, as demonstrated by the space-time spectral analysis of Fig. 6a, b. Indeed, over 99% of the pulse energy is carried by forward propagating waves (Fig. 6a2, b2). In addition, according to the standard criterion of nondiffraction for space-time wave packets[23,24], the STP spectrum has a broad distribution in the $k_z$-$\omega$ plane that indicates diffractive propagation. On the other hand, the NDSTP spectrum is confined in a conical section by a plane perpendicular to $k_z$-$\omega$ plane, which implies a close to uniform constant group velocity $v_g = \partial\omega/(\partial k_z) = c$ (the slope of the plane) along the $z$-axis and thus diffraction is almost completely absent. We note that while exact nondiffracting behavior corresponds to an infinitely thin line, NDSTPs approximate such behaviour closely, see details in Supplementary Note 4.

The generation of NDSTPs will involve addressing two main challenges: (1) The $\omega - k$ spectrum is distributed along a thin line of a conic section on the light cone, in order to fulfill the nondiffraction criterion; (2) The $\omega - k$ spectrum should include a null region at the center ($k_r = 0$), induced by the cylindrical polarization and vector singularity of the pulse. The former challenge can be addressed by realizing that the NDSTP pulse has a simple spectral structure in $\omega - k$ space as indicated by Fig. 6. Indeed, recently, photonic crystal slabs (PCS) consisting of 2D hole arrays were used to control the $\omega - k$ spectrum of emitted light in order to generate space-time bullet pulses[61]. Cylindrically polarized light fields have been generated by PCS of specific symmetry groups[62]. Hence, we argue that controlling the geometry and symmetry of PCS will allow to generate NDSTPs. Alternatively, the recent method of generating picosecond-level 3D nondiffracting wavepackets using transformation optics can be considered[63]. Such an approach would need to be extended to cover the broad bandwidth of NDSTPs and allow control of vector polarization.

Of particular importance here is the relation between the size of the generating metasurface (aperture) and the distance over which the generated pulses propagate without diffraction. Our numerical study shows that an aperture of $400\lambda$ is sufficient for nondiffracting propagation (within 1% of the pulse FWHM) over a distance of $10^5\lambda$ (see Supplementary Note 4). Moreover, in contrast to the well-known Bessel beams, NDSTPs are finite energy solutions to Maxwell's equations. Thus, we argue that the NDSTPs are less sensitive to the metasurface size and thus their practical implementations remain closer to their ideal form.

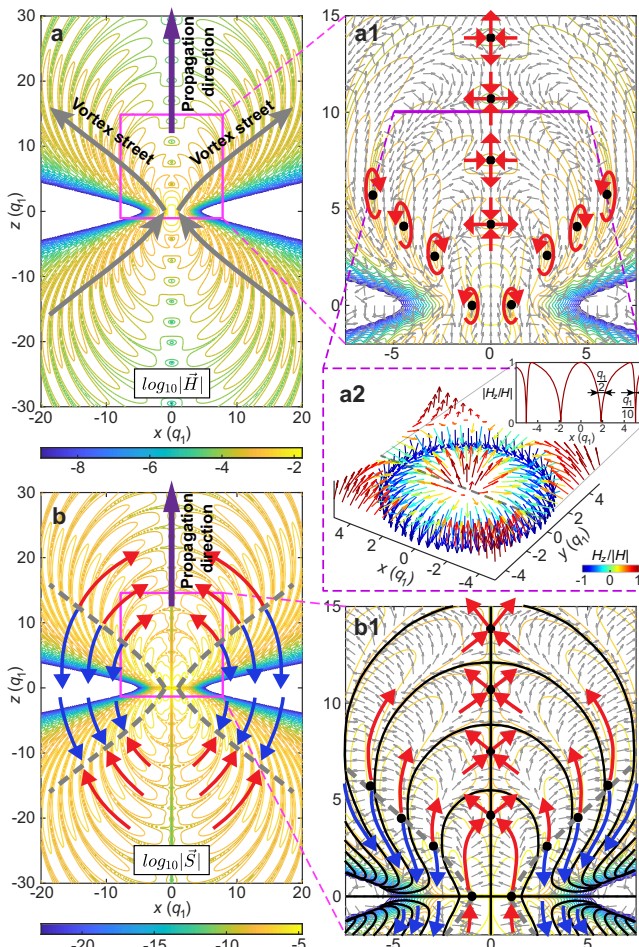

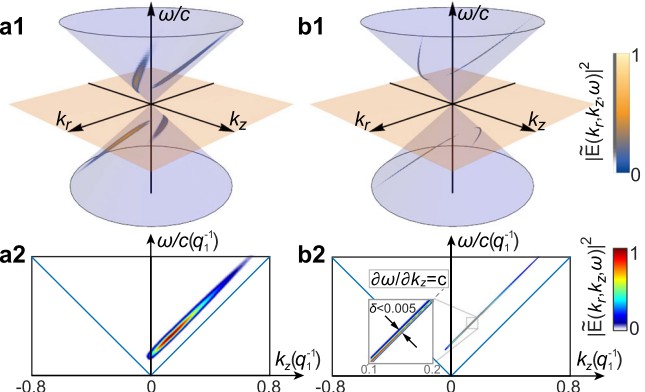

**Fig. 6 | Spectral analysis.** Plane wave spectra of an STP (**a1**) and a NDSTP (**b1**), and corresponding projections into the $k_z$-$\omega/c$ plane (**a2**, **b2**). The light cone is represented by the blue surface in (**a1**, **b1**) and the blue lines in (**a2**, **b2**). The parameters of the STP and NDSTP are the same as the ones in Fig. 3a, b).

**Fig. 4 | Topological electromagnetic and energy flow structures.** Magnetic field (**a**) and Poynting vector (**b**) distributions of a NDSTP with $q_2 = 100$ and $q_3 = 1$ at $t = 0$. The contours show the logarithm of the modulus of the corresponding vectors. **a1** A zoom-in of the magnetic field with the arrow plot showing the vector distribution, the black dots referring to the singularities, and surrounding red arrows marking the type of the vector singularities (vortex and saddle) serving as guide to the eye. **a2** Arrow plot showing a skyrmionic structure of magnetic field in the transverse ($x$-$y$) plane at $z = 10$, the insert highlights the deep-subwavelength features of the field vector reversal regions in the skyrmion texture. **b1** A zoom-in of the Poynting vector field, where the arrow plot corresponds to the vector distribution, solid lines and dots mark the zeros of the Poynting vector, and red and blue arrows indicate areas with forward and backward energy flow, respectively. Unit of length: $q_1$.

In conclusion, we demonstrate NDSTPs with robust topological field structures including fractal-like singularities, skyrmions, vortex rings, and energy backflow. In contrast to prior studies, the topological features of NDSTPs can stably propagate over an arbitrarily long distance. Our results provide a playground for the study of the propagation dynamics of electromagnetic skyrmions, as well as their interactions with matter, in particular in complex media with nonlinearity, anisotropy, or chirality. Due to its propagation-robust topology, the NDSTP acts as a spectacular display of staggered electromagnetic vortex dipole arrays with stable propagation, evocative of the classic KVS. The optical KVSs in NDSTPs unveil intriguing analogies between fluid transport and the flow of energy in structured light. The robust topological structure of NDSTPs that remains invariant upon propagation could be used for long-distance information transfer encoded in the topological features of the pulses with implications for telecommunications, remote sensing, and LIDAR. The toroidal field configuration of NDSTPs could be of interest in the spectroscopy of toroidal excitations in matter[33,64,65]. Finally, the deeply subwavelength singularities of NDSTPs can be employed for applications in metrology, where such topological features have been shown to lead to orders of magnitude improvement in precision and resolution[15].

## Data availability

The data from this paper can be obtained from the University of Southampton ePrints research repository at https://doi.org/10.5258/SOTON/D3058.

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

## Acknowledgements

The authors acknowledge the supports of the Singapore Ministry of Education (MOE) (MOE2016-T3-1-006), the UKs Engineering and Physical Sciences Research Council (grant EP/M009122/1, Funder Id: 10.13039/501100000266), the European Research Council (Advanced grant FLEET-786851, Funder Id: https://doi.org/10.13039/501100000781), and the Defense Advanced Research Projects Agency (DARPA) under the Nascent Light Matter Interactions program. Y.S. thanks the support of a start grant from Nanyang Technological University and Singapore MOE AcRF Tier 1 grant (RG157/23).

## Author contributions

Y.S. conceived the idea, performed the theory and simulations, and wrote the article. N.P. and N.I.Z. contributed to data analysis and interpretation and supervised the project. All authors contributed to the revisions of the paper.

## Competing interests

The authors declare no competing interests.
