## [Peer Review File · Nature Communications]

Nondiffracting supertoroidal pulses and optical “Kármán vortex streets”Reviewer #1 (Remarks to the Author):

This work focuses on an emerging research direction - the toroidal electrodynamics, which is motivated by the authors' group and attracts rising attentions. I found this work very impressive that it is demonstrated that the supertoroidal structures of light can be nondiffracting, and the topological structures can persist upon propagation over arbitrarily long distances. The supertoroidal is a novel topological structure recently proposed by the group, which promises many interesting properties. The new work here is the key step to push the supertoroidal into practical usage. This article is well written, and figures are well prepared and very pretty. I can tell the authors did major revisions and now the article was much improved than the arxiv version. Therefore, I believe this work opens a new multidisciplinary field of topological optics hydrodynamics simulation, and I can highly support this work to be published in NC.

Here are some revision suggestions:

(1) Although the behaviors are similar, the NDSTP pulses still have some fundamental difference with the real vortex streets. The NDSTP pulses exist in linear electromagnetic fields but the real vortex streets are in nonlinear fluid. Maybe it is more suitable to say "nondiffracting supertoroidal pulses analogue to optical vortex streets" in the title, rather than using colon. Also, another big difference is that the hydrodynamic vortex street is always a 2D model, but the new optical vortex streets are 3D, and the vortex is 3D vortex rings, it should be better to explain this clearly, and highlight it is vortex-ring street.

(2) The authors discussed the generation scheme, it is good. But the authors claim that the NDSTP can be generated by similar metasurface convertor as that to generate prior toroidal pulses. "We propose that NDSTPs can be generated by using a similar metasurface convertor to impart the required space-time correlations as indicated by Fig. 6" This is not very convincing. The real-space metasurface used to generate toroidal pulse is by tuning the radial gradient to fit spectral profile. However, the NDSTP is not only related the profile but also the thickness of the spectrum. The PCS is possible to do by k-space modulation. But for metasurface method the author introduced, I cannot find where is the addition parameter we can use to control this, which seems impossible. I suggest the authors to modify the statement if they agree.

(3) The authors may need to compare with Abouraddy's space-time wave packet, which can already control 3D space-time structures as well as 3D vector patterns. Can that be used to generate supertoroidal pulses? It is expected to be discussed.

(4) Table I should be Classification of supertoroidal pulses, not toroidal pulses?

(5) There are some typos and format issues in References, the formats of some journal names are not unified, e.g. some "Optical letters", some "Optical Letters"

Reviewer #2 (Remarks to the Author):

This paper reports on a novel family of mathematical solutions of Maxwell equations that correspond to special pulsed structures named by the authors "nondiffracting supertoroidal light pulses".

These are interesting structures, in principle, for which bundles of the field lines describe toroidal topologies. The authors of this paper have already published a few papers on related subfamilies of these same structures. The main novelty of this last manuscript appears to be the (almost) "non-diffracting" nature of these solutions for a proper choice of certain parameters. The authors acknowledge that exact non-diffraction is impossible, as exactly non-diffracting solutions are associated with an infinite total energy or power, as in Bessel beams. However, they argue that one can approximate these ideal solutions, thus obtaining realistic solutions that diffract very weakly for a fairly long length, similarly to the case of Gaussian-Bessel beams.

The manuscript stops short of rigorously proving that these new pulses are nondiffracting. The theoretical analysis, detailed in the supplementary material, employs approximations, rendering it an approximate argument only. Despite this, the theoretical discussion and accompanying numerical

calculations offer persuasive evidence of the result. The exact solutions also seem to include a minor fraction of counterpropagating waves, a hallmark of exact yet non-practical solutions (they would need to be launched from both directions). For the parameter range studied, this fraction is however small enough to allow one to assume that a similar behavior would be seen in waves without this counterpropagating component. Additionally, the paper seems to lack a formal proof that these nondiffracting solutions possess finite total energy, a necessary condition to avoid them being mere idealizations.

The research does not include experimental validation of the theoretical results, leaving the practical applicability of the approach unverified. Furthermore, the applications suggested for these specific nondiffracting toroidal pulses are rather generic and their practical value uncertain.

It is worth noting that such electromagnetic toroidal and skyrmionic structures might not be as exotic as presumed. Based on intuitive understanding, they could naturally arise in any cylindrical-symmetric geometry from the solenoidal nature of free-space electromagnetic fields and their alternating orientation in propagating waves. It seems likely that any finite cylindrically-symmetric optical beam would naturally form a sequence of alternating toroids, as further revealed by the alternating skyrmionic numbers reported in the manuscript (e.g., Figure 4).

Overall, although the reported work is certainly interesting for an audience of specialists, the degree of innovation and broad interest of its results is in my opinion insufficient for publication in a high-selectivity journal such as Nature Communications. Both the supertoroidal structures themselves (already published by the same group in 2021 in Nature Communications, see Ref. 39 of the manuscript; the first experiments have been published by the same group in Nature Photonics in 2022) and the existence of quasi-non-diffracting solutions are not novel concepts. In addition, the reported "vortex street" structure is also not unprecedented and not so striking.

In conclusion, this paper offers a theoretical exploration of a set of mathematical solutions to Maxwell's equations, describing quasi-nondiffracting pulses with toroidal field line topology. Although it may interest specialists in this field, I believe it does not meet the criteria for publication in Nature Communications.

Reviewer #3 (Remarks to the Author):

Based on the electromagnetic solution of focused few cycle electromagnetic waves proposed by Ziolkowski, the authors discovered a new regime of super toroidal pulses that are non-diffractive. Through detailed numerical study, the authors showed that the non-diffracting super toroidal (NDSTP) wave not only inherits the topological features of STPs, but also exhibits new topological features such as Karman vortex streets. The paper shows interesting physics and is well organized. In particular, the non-diffracting feature could lead to important applications in optical communications. I am happy to recommend its publication in Nature Communications once the following minor points are addressed.

My understanding is that since the NDSTPs carry finite energy, they cannot be strictly non-diffractive. While the proof of non-diffraction in the supplementary materials relies on some approximations (Eq. S23 and S24). With Gaussian beams, the product of the divergence angle and the waist size is a constant. I wonder if similar relation can also be obtained for a given central wavelength of the pulse.

In the paper the authors showed that "an aperture of 400λ is sufficient for non-diffracting propagation (within 1% of pulse FWHM) over a distance of $10^5 \lambda$ ". For practical applications this aperture size may be too large. The authors should provide more discussions on whether a reasonably good non-diffracting performance can still be obtained with a significantly smaller aperture size.

Comments from the reviewers:

Reviewer #1 (Comments to the Author):

“ This work focuses on an emerging research direction - the toroidal electrodynamics, which is motivated by the authors’ group and attracts rising attentions. I found this work very impressive that it is demonstrated that the supertoroidal structures of light can be nondiffracting, and the topological structures can persist upon propagation over arbitrarily long distances. The supertoroidal is a novel topological structure recently proposed by the group, which promises many interesting properties. The new work here is the key step to push the supertoroidal into practical usage. This article is well written, and figures are well prepared and very pretty. I can tell the authors did major revisions and now the article was much improved than the arxiv version. Therefore, I believe this work opens a new multidisciplinary field of topological optics hydrodynamics simulation, and I can highly support this work to be published in NC.”

Response: We thank the referee for their positive comments.

“ Here are some revision suggestions:

(1) Although the behaviors are similar, the NDSTP pulses still have some fundamental difference with the real vortex streets. The NDSTP pulses exist in linear electromagnetic fields but the real vortex streets are in nonlinear fluid. Maybe it is more suitable to say “nondiffracting supertoroidal pulses analogue to optical vortex streets” in the title, rather than using colon. Also, another big difference is that the hydrodynamic vortex street is always a 2D model, but the new optical vortex streets are 3D, and the vortex is 3D vortex rings, it should be better to explain this clearly, and highlight it is vortex-ring street.”

Response: Following the reviewer’s recommendation, we have revised the title as “Nondiffracting supertoroidal pulses: optical analogs of Kármán vortex streets”. We also highlight the differences between the 2D hydrodynamic and the 3D electromagnetic vortex streets in page 5, right column.

“ (2) The authors discussed the generation scheme, it is good. But the authors claim that the NDSTP can be generated by similar metasurface convertor as that to generate prior toroidal pulses. “We propose that NDSTPs can be generated by using a similar metasurface convertor to impart the required space-time correlations as indicated by Fig. 6” This is not very convincing. The real-space metasurface used to generate toroidal pulse is by tuning the radial gradient to fit spectral profile. However, the NDSTP is not only related the profile but also the thickness of the spectrum. The PCS is possible to do by k-space modulation. But for metasurface method the author introduced, I cannot find where is the addition parameter we can use to control this, which seems impossible. I suggest the authors to modify the statement if they agree.”

Response: We agree with the referee. Indeed, k-space modulated photonic crystal slabs are a simpler and more elegant solution to the problem of generating supertoroidal pulses. We have revised our manuscript accordingly (see 1st paragraph of the Discussion section, bottom of page 5 & top of page 6).

“ (3) The authors may need to compare with Abouraddy’s space-time wave packet, which can already control 3D space-time structures as well as 3D vector patterns. Can that be used to generate supertoroidal pulses? It is expected to be discussed.”

Response: We thank the reviewer for their suggestion. Abouraddy's approach may be used to generate NDSTPs. However, it will need to be extended in order to accommodate the broad bandwidth of the NDSTPs and provide at the same time vector polarization control. We now cite Abouraddy's recent work on 3D space-time wave packet, see Ref [62], and discuss its relevance to NDSTP generation at the end of the 1st paragraph of the Discussion section (top of page 6).

“ (4) Table I should be Classification of supertoroidal pulses, not toroidal pulses?”

Response: We have corrected this typo.

“ (5) There are some typos and format issues in References, the formats of some journal names are not unified, e.g. some “Optical letters”, some “Optical Letters”.”

Response: We have corrected the formatting and typos in References.

Reviewer #2 (Comments to the Author):

1. *“ This paper reports on a novel family of mathematical solutions of Maxwell equations that correspond to special pulsed structures named by the authors “nondiffracting supertoroidal light pulses”.*

These are interesting structures, in principle, for which bundles of the field lines describe toroidal topologies. The authors of this paper have already published a few papers on related subfamilies of these same structures. The main novelty of this last manuscript appears to be the (almost) “non-diffracting” nature of these solutions for a proper choice of certain parameters. The authors acknowledge that exact non-diffraction is impossible, as exactly non-diffracting solutions are associated with an infinite total energy or power, as in Bessel beams. However, they argue that one can approximate these ideal solutions, thus obtaining realistic solutions that diffract very weakly for a fairly long length, similarly to the case of Gaussian-Bessel beams.”

Response: We thank the referee for the positive comments. We note that the novelty of the manuscript is not limited to the non-diffracting nature of the pulses, but rather it involves the robustness of the topological features of the pulse upon propagation. We discuss this issue in more detail in our response to comment #5 of the same reviewer.

2. *“ The manuscript stops short of rigorously proving that these new pulses are nondiffracting. The theoretical analysis, detailed in the supplementary material, employs approximations, rendering it an approximate argument only. Despite this, the theoretical discussion and accompanying numerical calculations offer persuasive evidence of the result. The exact solutions also seem to include a minor fraction of counterpropagating waves, a hallmark of exact yet non-practical solutions (they would need to be launched from both directions). For the parameter range studied, this fraction is however small enough to allow one to assume that a similar behavior would be seen in waves without this counterpropagating component. Additionally, the paper seems to lack a formal proof that these nondiffracting solutions possess finite total energy, a necessary condition to avoid them being mere idealizations.”*

Response: A formal proof of the finite energy nature of the NDSTPs can be found in the

early work by Ziolkowski (ref. [45]) provided for the general class of pulses of which NDSTPs form a special case. This is now mentioned at the bottom of page 1, right column.

3. *“ The research does not include experimental validation of the theoretical results, leaving the practical applicability of the approach unverified. Furthermore, the applications suggested for these specific nondiffracting toroidal pulses are rather generic and their practical value uncertain.”*

Response: Our work introduces NDSTPs and focuses on the theoretical study of their propagation and topological properties. As such, experimentally verifying these theoretical predictions and demonstrating the practicality of the proposed applications lies beyond the scope of the manuscript. To address the reviewer’s concerns, we have expanded the discussion of NDSTP generation at the end of page 5 & top of page 6 and the discussion of potential applications in the concluding paragraph of the paper.

4. *“ It is worth noting that such electromagnetic toroidal and skyrmionic structures might not be as exotic as presumed. Based on intuitive understanding, they could naturally arise in any cylindrical-symmetric geometry from the solenoidal nature of free-space electromagnetic fields and their alternating orientation in propagating waves. It seems likely that any finite cylindrically-symmetric optical beam would naturally form a sequence of alternating toroids, as further revealed by the alternating skyrmionic numbers reported in the manuscript (e.g., Figure 4).”*

Response: We agree with the referee in that cylindrically-symmetric vector polarized beams will exhibit elements of toroidal and skyrmionic topology, see for instance refs [Phys. Rev. A 102, 053513 (2020); Opt. Lett. 46, 3737–3740 (2021); ACS Photonics 9, 296–303 (2022)]. In these instances, the skyrmionic field patterns are observed in the steady-state, continuous wave regime. In contrast, NDSTPs exhibit propagating spatiotemporal topological field configurations. Importantly, the space-time non-separability of NDSTPs leads to propagation without diffraction, fractal-like topological structure, and behaviour analogous to Karman vortex streets. This discussion is now included in page 1, bottom of left column.

5. *“ Overall, although the reported work is certainly interesting for an audience of specialists, the degree of innovation and broad interest of its results is in my opinion insufficient for publication in a high-selectivity journal such as Nature Communications. Both the supertoroidal structures themselves (already published by the same group in 2021 in Nature Communications, see Ref. 39 of the manuscript; the first experiments have been published by the same group in Nature Photonics in 2022) and the existence of quasi-non-diffracting solutions are not novel concepts.”*

Response: We respectfully disagree with the referee. Our prior work introducing supertoroidal pulses exhibiting skyrmion-like topology in free-space (Nat. Comm. 2021) together with the experimental demonstration of the elementary toroidal pulse (Nat. Photon. 2022) led to an explosion of activity in this new field. However, the skyrmion-like topology was only available to study at localized regions around the focus, preventing further studies of the propagation dynamics of such skyrmionic light fields. In contrast, our current manuscript demonstrates:

- (1) Non-diffracting space-time nonseparable pulses, in essence, electromagnetic “quasi-particles” of skyrmionic topology.
- (2) Whereas numerous examples of localized (non-propagating) electromagnetic skyrmions or propagating waves that exhibit skyrmionic topology in a limited region in space (e.g. at the focus), the nondiffracting supertoroidal pulses retain their skyrmionic nature along arbitrary long propagation trajectories.
- (3) In contrast to already identified nondiffracting Bessel Beams, that are single-frequency beams, and X-waves, the nondiffracting supertoroidal pulses represent the first example of electromagnetic wave of toroidal symmetry with a broad continuous spectrum that is non-diffracting.

Therefore, this work is a substantial step forward in electrodynamics, which allows for the first time the study of the propagation dynamics of electromagnetic skyrmion-like fields, but also enables applications, such as information transfer schemes based on robust transport of topological properties.

This discussion is included at the end of the introduction section.

6. *“ In addition, the reported “vortex street” structure is also not unprecedented and not so striking.”*

Response: We are not aware of any prior work on 3D propagating optical vortex streets. Indeed, earlier works (Refs [42,43]) on this topic involved 2D vortex streets formed by a phase singularity in continue-wave light field, not a propagating pulse. Our work is the first demonstration of a “vortex street” in structured pulses with staggered vortex arrays stably propagating forward. This is now emphasized in page 5, right column of the revised manuscript.

7. *“In conclusion, this paper offers a theoretical exploration of a set of mathematical solutions to Maxwell's equations, describing quasi-nondiffracting pulses with toroidal field line topology. Although it may interest specialists in this field, I believe it does not meet the criteria for publication in Nature Communications.”*

Response: We argue that the introduction of non-diffracting topological pulses is of great importance to the fields of topological optics and structured light and will be of interest to the broader research community working on fundamental and applied optical science. We hope that our responses to the reviewer’s comments, in particular comment 5., make a convincing case to this end.

Reviewer #3 (Comments to the Author):

“ Based on the electromagnetic solution of focused few cycle electromagnetic waves proposed by Ziolkowski, the authors discovered a new regime of super toroidal pulses that are non-diffractive. Through detailed numerical study, the authors showed that the non-diffracting super toroidal (NDSTP) wave not only inherits the topological features of STPs, but also exhibits new topological features such as Karman vortex streets. The paper shows interesting physics and is well organized. In particular, the non-diffracting feature could lead to important applications in optical communications. I am happy to recommend its publication in Nature Communications once the following minor points are addressed.”

Response: We thank the referee for the very positive comments.

“ My understanding is that since the NDSTPs carry finite energy, they cannot be strictly non-diffractive. While the proof of non-diffraction in the supplementary materials relies on some approximations (Eq. S23 and S24). With Gaussian beams, the product of the divergence angle and the waist size is a constant. I wonder if similar relation can also be obtained for a given central wavelength of the pulse. ”

Response: We thank the reviewer for their suggestion. Indeed, all physical nondiffracting light beams depend on certain assumptions due to the finite energy requirement. For Gaussian beams, the product of the divergence angle and the waist size is a constant ($\theta w_0 = \lambda/\pi$), usually called the beam parameter. For the fundamental toroidal pulse, there is a similar relationship between the divergence and waist size, i.e. the beam size $w_0 = \sqrt{q_1 z_0}$, where q_1 is related to the central wavelength, $z_0 = q_2/2$ is the Rayleigh range. For nondiffracting Bessel-Gauss beams, this parameter can be small (but not identically zero). For NDSTPs, we demonstrate numerically that with decreasing parameter q_3 , the Rayleigh range increases exponentially, or equivalently the beam parameter -- product of the divergence angle and the waist size is rapidly decreasing (see figure below). We are not aware of an analytical expression for the relation between divergence angle and waist size.

We have added the discussion in Supplementary Note 2, highlighted by blue.

“ In the paper the authors showed that “an aperture of 400 lambda is sufficient for non-diffracting propagation (within 1% of pulse FWHM) over a distance of 10^5 lambda”. For practical applications this aperture size may be too large. The authors should provide more discussions on whether a reasonably good non-diffracting performance can still be obtained with a significantly smaller aperture size.”

Response: Following the reviewer’s recommendation, we considered apertures with diameters from 0.5 mm (~400 λ) down to 40 μ m (~32 λ) and demonstrate that the nondiffracting propagation distance varies between 150 mm (1.2 x 10⁵ λ) and 6 mm (4800 λ). These values are comparable if not superior to those reported in the literature for different types of non-diffracting beams [Nat. Photonics 11, 733–740 (2017); Sci. Rep. 10, 21981 (2020); Appl. Phys. Lett. 110, 114102 (2017)]. This discussion is now included at the end of the Supplementary material.

Reviewer #1 (Remarks to the Author):

The authors have fully addressed my comments and the manuscript is largely improved now. I also believe the authors made very convincing responses to Referee #2. The nondiffraction supported 3D vortex street should be a very meaningful and new topology of light. Therefore I highly recommend publication of current version in NC.

Reviewer #2 (Remarks to the Author):

I have carefully reviewed the revised manuscript, alongside the authors' responses to the comments made by the reviewers. The revisions have indeed improved the manuscript, effectively addressing all minor concerns previously raised. However, my principal critique regarding the manuscript's claimed degree of innovation and its suitability for publication in Nature Communications remains unchanged.

While I acknowledge that the content is original and introduces new ideas, I maintain that these contributions are fundamentally incremental. They primarily extend from the integration of well-established concepts, the combination of which seemed relatively obvious. Therefore, despite the improvements, my assessment of the manuscript's fit for Nature Communications based on its level of innovation stands.

Reviewer #3 (Remarks to the Author):

The authors have successfully addressed all the questions raised in my previous report. I only have one last minor question, which I hope the authors can elucidate. Other than that, I am happy to recommend its publication in Nature Communications.

In their manuscript, the authors mentioned that the parameter q_1 is approximately 0.24λ when q_3 is infinity. I am curious about how q_1 varies when q_3 becomes finite. In Fig. 2c-f, it appears that all parameters are normalized by q_1 . I am wondering whether q_1 remains the same in these plots. If q_1 does remain constant, it is quite surprising to observe that the smaller the beam, the smaller the divergence angle, which contradicts intuition. Conversely, if q_1 varies in these plots, the authors may want to consider normalizing all parameters by the wavelength, instead of q_1 .

Reviewer #1 (Comments to the Author):

“The authors have fully addressed my comments and the manuscript is largely improved now. I also believe the authors made very convincing responses to Referee #2. The nondiffraction supported 3D vortex street should be a very meaningful and new topology of light. Therefore I highly recommend publication of current version in NC.”

Response: We thank you for your positive comments and supports.

Reviewer #2 (Comments to the Author):

“I have carefully reviewed the revised manuscript, alongside the authors' responses to the comments made by the reviewers. The revisions have indeed improved the manuscript, effectively addressing all minor concerns previously raised. However, my principal critique regarding the manuscript's claimed degree of innovation and its suitability for publication in Nature Communications remains unchanged.

While I acknowledge that the content is original and introduces new ideas, I maintain that these contributions are fundamentally incremental. They primarily extend from the integration of well-established concepts, the combination of which seemed relatively obvious. Therefore, despite the improvements, my assessment of the manuscript's fit for Nature Communications based on its level of innovation stands.”

Response: We thank you for the positive comments and acknowledgement of the improvement of our article. We argue that the introduction of non-diffracting topological pulses is of great importance to the fields of topological optics and structured light and will be of interest to the broader research community working on fundamental and applied optical science.

Reviewer #3 (Comments to the Author):

“The authors have successfully addressed all the questions raised in my previous report. I only have one last minor question, which I hope the authors can elucidate. Other than that, I am happy to recommend its publication in Nature Communications.”

Response: We thank you for your positive comments and supports.

“In their manuscript, the authors mentioned that the parameter q_1 is approximately 0.24λ when q_3 is infinity. I am curious about how q_1 varies when q_3 becomes finite. In Fig. 2c-f, it appears that all parameters are normalized by q_1 . I am wondering whether q_1 remains the same in these plots. If q_1 does remain constant, it is quite surprising to observe that the smaller the beam, the smaller the divergence angle, which contradicts intuition. Conversely, if q_1 varies in these plots, the authors may want to consider normalizing all parameters by the wavelength, instead of q_1 .”

Response: In Fig. 2c-f, q_1 is equal to unity for all cases. We also note that q_1 is independent of q_3 and thus we would not expect a change in the former when the latter varies. The counterintuitive decrease in beam waist with decrease of divergence angle can be attributed to the presence of “long thick wings” at the peripheral area of the pulse.

This discussion is now included in the description of Fig.2. We emphasize the relation between wavelength and q_1 in Page-2 the caption of Fig.2 and Page-3 right-middle part of main text.